# Differential Oligomerization of Alpha versus Beta Amino Acids and Hydroxy Acids in Abiotic Proto-Peptide Synthesis Reactions

**DOI:** 10.3390/life12020265

**Published:** 2022-02-10

**Authors:** Moran Frenkel-Pinter, Kaitlin C. Jacobson, Jonathan Eskew-Martin, Jay G. Forsythe, Martha A. Grover, Loren Dean Williams, Nicholas V. Hud

**Affiliations:** 1NSF-NASA Center for Chemical Evolution, Georgia Institute of Technology, Atlanta, GA 30332, USA; moran.fp@mail.huji.ac.il (M.F.-P.); kcj@gatech.edu (K.C.J.); eskewmartinjc@g.cofc.edu (J.E.-M.); forsythejg@cofc.edu (J.G.F.); martha.grover@chbe.gatech.edu (M.A.G.); 2School of Chemistry & Biochemistry, Georgia Institute of Technology, Atlanta, GA 30332, USA; 3Institute of Chemistry, The Hebrew University of Jerusalem, Jerusalem 91904, Israel; 4Department of Chemistry and Biochemistry, College of Charleston, Charleston, SC 29424, USA; 5School of Chemical & Biomolecular Engineering, Georgia Institute of Technology, Atlanta, GA 30332, USA

**Keywords:** prebiotic chemistry, condensation dehydration, peptide evolution, chemical evolution, depsipeptides

## Abstract

The origin of biopolymers is a central question in origins of life research. In extant life, proteins are coded linear polymers made of a fixed set of twenty alpha-_L_-amino acids. It is likely that the prebiotic forerunners of proteins, or protopeptides, were more heterogenous polymers with a greater diversity of building blocks and linkage stereochemistry. To investigate a possible chemical selection for alpha versus beta amino acids in abiotic polymerization reactions, we subjected mixtures of alpha and beta hydroxy and amino acids to single-step dry-down or wet-dry cycling conditions. The resulting model protopeptide mixtures were analyzed by a variety of analytical techniques, including mass spectrometry and NMR spectroscopy. We observed that amino acids typically exhibited a higher extent of polymerization in reactions that also contained alpha hydroxy acids over beta hydroxy acids, whereas the extent of polymerization by beta amino acids was higher compared to their alpha amino acid analogs. Our results suggest that a variety of heterogenous protopeptide backbones existed during the prebiotic epoch, and that selection towards alpha backbones occurred later as a result of polymer evolution.

## 1. Introduction

All living organisms utilize the same limited sets of small molecules as building blocks of polynucleotides, polypeptides, and polysaccharides [1]. Reasonable pathways for the origins of some building blocks, especially amino acids, have been proposed [2,3,4,5,6,7,8,9,10,11,12,13,14,15,16,17,18]. It is highly probable that amino acids were present on the prebiotic Earth, via both production in situ [3,4,6,7,8] and as well as by exogenous delivery [5,8,12,16]. It is possible that, around four billion years ago, a creative phase of pre-Darwinian chemical evolution pared down the number of potential building blocks to form oligomers and polymers with levels of functionally required for Darwinian processes [15,19,20,21,22,23,24]. Numerous models have been proposed to explain various chemical and physical aspects of the origins of biopolymers [15,18,22,25,26,27,28,29,30,31,32,33,34,35,36,37,38,39,40]. The chemistry of water and the condensation-dehydration of building blocks into chimeric metabolites, oligomers and polymers are important features of these models. 

Here, we experimentally investigate chemical processes that might have contributed to the selection of proteinaceous amino acids and of the polypeptide backbone. We focus on comparing reactivities of alpha amino acids versus beta amino acids in model abiotic polymerization reactions. Coded protein, a foundation of biological chemistry, is a linear polymer built from 19 alpha-_L_-amino acids and achiral glycine, with a cohesive backbone that self-assembles into α-helices and β-sheets via paired hydrogen-bond donors and acceptors [22]. The prebiotic milieu would have been rich in amino acids varying in chirality, side-chain functional groups, and isomeric forms (e.g., alpha-, beta-, and gamma-amino acids) [8,16,25,41,42,43]. It is possible that some degree of refinement of heterogenous, prebiotic, noncoded protopeptides during the transition to biotic coded proteins was a result of the intrinsic reactivities of the monomers in oligomerization reactions, a refinement contributing to pre-Darwinian chemical evolution.

Depsipeptides, which contain mixtures of amide and ester linkages, are plausible protopeptide candidates [25,32,44,45,46,47]. Hydroxy acids readily form esters when dried, and via ester-amide exchange, act as catalysts for the generation of amide bonds [25,32,44,45]. Wet-dry cycling can drive the oscillation between condensation at low water activity and hydrolysis in high water activity, based on Le Chatelier’s principle [48]. Depsipeptides are progressively enriched with amide bonds over ester bonds with repeated wet-dry cycles due to the differential hydrolysis rates of these two distinct types of linkages [45]. Peptide bond formation by hydroxy-acid-mediated ester-amide exchange avoids formation of diketopiperazine (DKP), a kinetic trap that prevents peptide elongation [49,50,51]. Hydroxy acids are found alongside amino acids in meteorites and are produced in model prebiotic reactions [41,52,53], arguably making depsipeptides and hydroxy-acid-capped peptides more plausibly prebiotic than pure peptides.

Several reports have discussed the formation of peptides containing beta amino acids in model abiotic reactions [54,55,56,57,58,59,60,61,62,63]. For instance, it has been shown that β-alanine is polymerized into β-peptides during wet-dry cycling in the presence of glycerol and sodium biocarbonate [54]. In addition, polymerization was demonstrated with beta amino acids activated with 1-ethyl-3-(3-dimethylaminopropyl)carbodiimide (EDAC) [55,56] and by the methyl ester forms of beta amino acids [54]. Moreover, alpha amino acids have different inherent polymerization behaviors to analogous beta and gamma amino acids [58,59]. For example, Liu and Orgel showed that β-glutamic acid polymerizes more efficiently than its alpha analog in the presence of activating EDAC [55]. On the other hand, Rode showed that alpha amino acids polymerize faster than their beta and gamma analogs in salt-induced peptide formation reactions [58]. More recently, Wu et al. demonstrated that under moderately acidic conditions (pH 3), glycine has higher reactivity than β-alanine in the presence of methyl isonitrile as an activating agent [59]. While prior investigations demonstrated that beta peptides can form abiotically, differences in reactivities of alpha amino acids versus beta amino acids in the absence of high-energy activating agents have not been investigated. Moreover, depsipeptide formation from beta hydroxy acids, and its comparison to depsipeptide formation from alpha hydroxy acids, has not been reported.

Here, we report a systematic comparison of the reactivities of alpha versus beta amino and hydroxy acids in depsipeptide polymerization reactions in single-step dry-down or wet-dry cycling reactions in the absence of high-energy activating agents. The depsipeptide products were analyzed by a variety of analytical methods, including matrix-assisted laser desorption/ionization–time-of-flight (MALDI-TOF) mass spectrometry, liquid chromatography–mass spectrometry (LC-MS), nuclear magnetic resonance spectroscopy (NMR), Fourier-transform infrared spectroscopy (FTIR), and high-performance liquid chromatography (HPLC). We found that the extent of polymerization of amino acids into depsipeptides is higher in the presence of alpha hydroxy acids than with beta hydroxy acids. Moreover, the extent of incorporation of beta amino acids into oligomers is greater than that of alpha amino acids. These results suggest that heterogenous depsipeptide and peptide backbones existed on the prebiotic Earth, and that the selection for alpha backbones did not arise solely from differences in chemical reactivities of the amino acids themselves, but came from alternative evolutionary mechanisms.

## 2. Materials and Methods

### 2.1. Materials

All chemicals and reagents were of analytical grade, including glycolic acid (Acros Organics, Thermo Fisher Scientific, Branchburg, NJ, USA, #15451-0250), L-lactic acid (TCI, Zwijndrecht, Belgium, #L0165), 3-hydroxy propanoic acid (Sigma-Aldrich, St. Louis, MO, USA, #792659), (S)-3-hydroxy butyric acid (Sigma-Aldrich, St. Louis, MO, USA, #54925), glycine (BioRad, Hercules, CA, USA, #161-0717), L-alanine (Alfa Aesar, Haverhill, MA, USA, #A15804), β-alanine (Sigma-Aldrich, St. Louis, MO, USA,, #05159), and (S)-3-amino butyric acid (Sigma-Aldrich, St. Louis, MO, USA, #757454). 

### 2.2. Synthesis of Peptide Standards

See supporting information.

### 2.3. Single-Step Dry-Down Reactions

For the formation of depsipeptides, aqueous solutions of hydroxy acids and amino acids at a 1:1 molar ratio (50 µmol each) were dried with caps open at 85 °C under unbuffered conditions (starting pH ~ 3–4) for one week. Control reactions were performed with either a hydroxy acid alone or an amino acid alone. Prior to analysis, the single-step dry-down reactions were resuspended in ultrapure water to give a 100 mM concentration, referring to the original amino acid and hydroxy acid concentration prior to incubation under evaporative conditions. The solutions were then vortexed and sonicated in ice before being diluted to the specified concentration for further analysis, as specified below. All product depsipeptide mixtures were fully soluble in water, except for the mixture containing lactic acid and alanine, for which only the supernatant has been analyzed. 

### 2.4. Wet-Dry Cycling 

For daily wet-dry cycling over a period of one week, aqueous solutions of hydroxy acids and amino acids at a 1:1 molar ratio (50 µmol each) were dried with the cap open for 18 h at 85 °C and then resuspended with 100 μL of ultrapure water. The solutions were vortexed and centrifuged, then immediately placed back at 85 °C with the cap on for 6 h. The caps were reopened again at the end of the 24-h period to start another wet-dry cycle. This was repeated for 7 cycles over the 7 days period. We did not perform control reactions involving amino acids in the absence of hydroxy acids in wet–dry cycling reactions because we did not expect significant differences in oligomerization in one-step dry-down reactions versus cycling reactions. For the extended cycling over a period of eight weeks, the reactions were first dried for 96 h, and then resuspended with ultrapure water to reconstitute 100 mM with respect the initial amino/hydroxy acid concentration. The samples were vortexed and centrifuged, and an aliquot was taken out after each cycle. The remaining sample was then placed back at 85 °C with closed caps for 72 h. At the end of the seven-day wet-dry cycle, the caps were reopened, and the next wet-dry cycle began. 

### 2.5. MALDI-TOF Analysis 

MALDI-TOF mass spectrometry was performed on a refurbished Voyager DE-STR instrument (JBI Scientific, Huntsville, TX, USA) equipped with a 337 nm nitrogen laser, which has a repetition rate of 20 Hz. All mass spectra were acquired in positive ion, reflector TOF mode. Other instrument settings were as follows: 20 kV, 75% extraction grid, 150 ns extraction delay, and *m*/*z* range 200–2000. Samples were initially 100 mM in acid monomer before polymerization and were diluted 1:10 (*v*/*v*) with 50% distilled water/50% acetonitrile solvent containing 0.1% trifluoroacetic acid. Next, 1 μL of 10 mg/mL 2,5-dihydroxybenzoic acid (DHB) matrix and 1 μL of diluted sample were spotted on the MALDI target and were cocrystallized at room temperature before analysis. Peak assignments correspond to either [M+H]^+^, [M+Na]^+^, or [M+K]^+^ ions. MALDI-MS data were processed using OriginPro software.

### 2.6. ESI-Mass Spectrometry

ESI-MS data were collected using an Agilent 6130 single quadrupole mass spectrometer with a capillary voltage of 3.0 kV and a source fragmentation voltage of 70 V (lab frame). Samples were diluted to 1 mM (referring to the original amino acid and hydroxy acid concentration) and directly infused into the mass spectrometer using the following parameters: Binary running solvents: 95% H_2_O, 5% acetonitrile with a flow rate of 0.5 mL/min. Injection volume: 5 μL with H_2_O needle wash. Path length: 0.6 cm. Scan range: 65–2000 *m*/*z*. MS data were processed using a suite of macros in Igor Pro-8.0 [33]. All peak assignments correspond to [M-H]^−^ ions. 

### 2.7. Liquid Chromatography-Mass Spectrometry 

LC-MS data were collected on an Agilent 1260 HPLC coupled to an Agilent 6130 single quadrupole mass spectrometer and an inline Agilent UV absorbance detector (210 nm) using 2.0-kV electrospray ionization (ESI) capillary voltage. Path length was 0.6 cm. Then, 100 mM samples (referring to the original amino acid and hydroxy acid concentration) were separated via a Kinetex XB-C18 column (150 × 2.1 mm, 2.6 μm particle size) under the same conditions described in the HPLC method. Eluted peaks were detected in negative-mode ESI-MS, scanning from 65–2000 *m*/*z* with 70 V source fragmentation voltage.

### 2.8. High-Performance Liquid Chromatography

HPLC analyses were conducted using an Agilent 1260 quaternary pump and autosampler with DAD UV-vis detector at 210 nm (Agilent Technologies, Santa Clara, CA, USA) under the following conditions: Path length 1.0 cm. Injection volume: 10 μL with H_2_O needle wash. Injection Speed: 100 μL/s. Further, 100 mM samples were separated by hydrophobicity on a Kinetex XB-C18 column (150 × 2.1 mm, 2.6 μm particle size) with a 0.3 mL/min flow rate and constant 25°C column temperature. Solvents: solvent gradient was as follows: (A) 0.1% formic acid in LC-MS-grade water, (B) LC-MS-grade acetonitrile. The method is as follows: 5 min 100% A, 0% B; 20 min ramp to 45% A, 55% B; 5 min ramp to 0% A, 100% B; 5 min 0% A, 100% B; 1 min ramp to 100% A, 0% B; 14 min 100% A, 0% B. The data were processed using a suite of macros in Igor Pro-8.0. Full HPLC spectra can be found in the Supporting Information.

### 2.9. NMR Spectroscopy

^1^H NMR Spectra were recorded on a Bruker Advance II-500 MHz (Billerica, MA, USA). In total, 25 μL of 100 mM samples (referring to the original amino acid and hydroxy acid concentrations) were lyophilized and then resuspended in 100 mM phosphate buffer in D_2_O (pH = 2.63) with 3-(trimethylsilyl)-1-propanesulfonic acid-d6 sodium salt (Sigma-Aldrich, St. Louis, MO, USA, #613150) as an internal standard (5 mM final proton concentration). A long relaxation delay of 15 s was selected to ensure quantitative integration of the resonances. The data were processed and plotted with TopSpin or MestReNova software packages. 

The overall conversion is the percent estimate of conversion of amino acid monomer into products. All conversions were calculated by comparing the integration of the free, nonamidated α- or β-protons of the reacted mixture to the integration of the unreacted stock mixture in ^1^H-NMR. 

For all monomer combinations that included glycine (i.e., glc+Gly, lac+Gly, hpa+Gly, and hba+Gly), the integration of the alpha protons at 3.72 ppm was used for calculations. Similarly, all calculations for combinations with L-alanine used the alpha proton at 3.95 ppm. For reactions containing β-alanine, the alpha protons at 2.79 ppm were selected for the combinations of glc+β-Ala, lac+β-Ala and hpa+β-Ala, while the beta protons at 3.27 ppm were selected for hba+β-Ala. For reactions containing β-aminobutyric acid, the beta proton at 3.74 ppm was selected for glc+β-Aba and hba+β-Aba, while the alpha proton at 2.74 ppm was selected for lac+β-Aba and hpa+β-Aba. 

### 2.10. FTIR Spectroscopy

IR spectra were analyzed on a Thermo Nicolet 4700 FTIR Spectrometer (Thermo Fisher Scientific, Waltham, MA, USA). Prior to analysis, 10 μL of the 100 mM samples were placed on Durapore^®^ hydrophobic PVDF Membranes with a pore size of 0.22 µm (Millipore Sigma, St. Louis, MO, USA, #GVHP04700) and allowed to dry. The membrane was then placed in an Attenuated Total Reflectance (ATR) sample chamber for analysis of the dried sample. Spectra were signal-averaged (16 scans per spectrum) and background-subtracted and ranged from 400 to 4000 cm^−1^.

### 2.11. Circular Dichroism

The CD spectra were collected at room temperature (25 °C) using a JASCO J-810 CD spectrometer. The samples were diluted to 1 mM amino acid or hydroxy acid (referring to the starting monomer concentration) in water, placed in a 1 mm cuvette, and then scanned from 260 nm to 190 nm with a resolution of 1 nm.

## 3. Results

### 3.1. Alpha and Beta Amino and Hydroxy Acids Exhibit a Different Extent of Polymerization in Single-Step Dry-Down Reactions

To examine whether alpha and beta amino acids and hydroxy acids exhibit differential extents of polymerization, we utilized a set of monomers composed of various simple hydroxy and amino acids (Figure 1). Four hydroxy acids were studied: glycolic acid (glc), lactic acid (lac), 3-hydroxypropionic acid (hpa), and 3-hydroxybutanoic acid (hba). glc and lac are alpha hydroxy acids, whereas hpa and hba are beta hydroxy acids. We also examined their corresponding amino acid analogs: glycine (Gly), alanine (Ala), β-alanine (β-Ala), and β-aminobutyric acid (β-Aba). Gly and Ala are alpha amino acids, whereas β-Ala and β-Aba are beta amino acids. This set of building blocks allowed us to explore whether differences between the alpha and beta monomer units modulate oligomerization. Methylated α- or β-carbons of the amino and hydroxy acids were included to test the effects of steric hindrance on the adjacent nucleophilic amine (in amino acids) or alcohol (in hydroxy acids) on the chemical reactivity of the various acids. Mixtures of a single amino acid with a single hydroxy acid at a 1:1 molar ratio were subjected to either single-step dry-down reactions or wet-dry cycling (6 h wet phase, 18 h dry phase) for one week at 85 °C under unbuffered, mildly acidic conditions (starting pH ~ 3–4).

MALDI-MS and LC-MS indicated that copolymers of hydroxy acids and amino acids were formed in all 16 combinations (Figure 2 and Appendix A). Depending on the composition of the starting mixture, oligomers with different backbones were formed: homo-alpha, homo-beta, and heterogenous backbones with both alpha and beta hydroxy acids and/or amino acids. The single-step dry-down reactions produced heterogenous oligomers of various lengths for the different mixtures (Figure 2 and Appendix A). For example, MS analysis indicated the formation of up to about 16-mers for lac+Ala and hpa+Ala, whereas shorter oligomers were observed for lac+β-Ala and hpa+β-Ala (up to 9-m) (Figure 2). Examples of depsipeptide products include 10lac5Ala and 5hpa4β-Ala, with each compositional species observed by LC-MS representing multiple possible sequence isomers. Overall, reactions containing alpha amino acids appeared to produce longer oligomers than reactions containing beta amino acids (Figure 2 and Appendix A). Notably, these observations are qualitative in nature since our MS analysis is not quantitative.

Reactions containing the four individual hydroxy acids in the absence of amino acids produce polyesters, as expected (Appendix A). Control reactions containing either Gly or Ala alone in the absence of hydroxy acids did not produce oligomers, also as expected (Appendix A). In contrast, reactions containing only β-Ala or β-Aba in the absence of hydroxy acids showed evidence of short peptides produced in small amounts (Appendix A). Confirmation of dimer formation in reactions with these beta amino acids by MS was complicated by the fact that trace amounts of β-Aba dimers and β-Ala dimers were detected in the unreacted stock solutions of these beta amino acids (Appendix A). However, reactions with β-Ala showed detectable amounts of trimers or tetramers, which were not observed in the unreacted stock solution, indicating that β-Ala can polymerize to some extent in the absence of hydroxy acids (Appendix A).

FTIR analysis of glc+Gly reaction products shows that single-step dry-down reactions induced shift from 1716 cm^−1^ to 1740 cm^−1^ in the C=O stretch, which is indicative of a shift from free carboxylic acids to the ester linkages that are associated with depsipeptide formation (Appendix A). Similar shifts are observed in the amide I and amide II regions for single-step dry-down reactions, supporting the formation of amide bonds (Appendix A). FTIR analyses of product mixtures for the other mixtures of hydroxy and amino acids supported the formation of depsipeptides in all cases, with evident shifts in the C=O band and in the amide regions (Appendix A). Circular dichroism (CD) analysis indicated shifts in the secondary structure of the depsipeptides following reactions under either single-step dry-down or wet-dry cycling conditions (Appendix A). The resulting depsipeptides typically exhibited a CD signature consistent with a random coil peptide conformation, with the exception of products from reactions involving mixtures of lac with Gly and lac with β-Ala, which displayed a spectrum similar to that associated with peptide β-sheet (Appendix A).

For a more quantitative analysis of product distribution, single-step dry-down reactions product mixtures were analyzed using hydrophobicity-based C18-HPLC (Figure 3 and Appendix A). Two possible heterodimers composed of a single hydroxy acid and a single amino acid can form in these reactions, with either amino acid or hydroxy acid on the C-terminus. However, in most product mixtures, one predominant heterodimer peak was evident that resulted from the condensation of a single hydroxy acid and a single amino acid. On the other hand, mixtures containing β-Aba typically showed two product peaks with the molecular mass corresponding to that of a heterodimer (i.e., a dimer composed of one β-Aba and one hydroxy acid). Further investigation using pure synthesized HO-terminated peptide standards showed that the predominant heterodimer peak is terminated with an amino acid on the C-terminus for all of the β-Aba reaction mixtures (Appendix A).

### 3.2. Differential Polymerization of Alpha versus Beta Amino Acids in Single-Step Dry-Down Reactions

The extent of polymerization by alpha and beta amino acids was also quantitated by ^1^H NMR spectroscopy. Amide and ester bond formation results in characteristic changes in amino acid proton chemical shifts [32], allowing the extent of incorporation of amino acids into oligomers to be determined by measuring the difference between the integrated intensity of the free, unreacted α-protons’ or β-protons’ resonances with the integrated intensity of these resonances prior to a single-step dry-down reaction (Figure 4 and Table 1). This analysis revealed that 98% of Gly and 96% of β-Ala were converted into products (Figure 4 and Table 1).

The extent of alpha and beta amino acid incorporation into products was typically higher in mixtures with alpha hydroxy acids than in mixtures with beta hydroxy acids. A comparison of product distributions for reactions of amino acids in all 16 combinations and under two reaction conditions (single-step dry-down reactions versus wet-dry cycling) is shown in Table 1 (NMR spectra are shown in Appendix A). Gly reacted extensively (98%) in the presence of glc and significantly less (29%) in the presence of hpa, which is the β-analog of glc. Similarly, 13% of Ala reacted in the presence of lac, but only 8% in the presence of hba, which is the β-analog of lac. The only exception to this general pattern was β-Aba, which showed 60% conversion into oligomers when mixed with glc but 72% when mixed with hpa, the β-analog of the glc. Steric hindrance of the amino and hydroxy acids can also play a decisive role in determining reactivity, as reactions with methylated analogs of the carbon adjacent to the nucleophilic amine (in amino acids) or alcohol (in hydroxy acids) resulted in less polymerization in comparison to nonmethylated analogs. For example, β-Ala exhibited 96% conversion into oligomers when mixed with glc but only 77% when mixed with lac, which is the methylated analog of glc. Similarly, when mixed with lac, 77% of β-Ala reacted, while only 33% conversion was observed for β-Aba, which is the β-methylated analog of β-Ala. We confirmed the reproducibility of these results with independent replica experiments (Appendix A).

### 3.3. Comparison of Single-Step Dry-Down Reactions versus Wet-Dry Cycling

To assess the effect of wet-dry cycling on amino acid polymerization, we compared the products of 16 different reactant combinations following either a single-step dry-down or wet-dry cycling (6 h wet phase, 18 h dry phase) for one week at 85 °C (Table 1). From a thermodynamic perspective, polymerization is favored under dry conditions while hydrolysis is favored in high water activity. In half of the reactions studied (8/16), a higher extent of polymerization was observed following a single-step dry-down reaction compared to wet-dry cycling (Figure 5). However, in some cases greater conversion was observed upon wet-dry cycling (Figure 5). For example, Ala reacted more extensively when subjected to wet-dry cycling than the single-step dry-down reactions when mixed with any of the hydroxy acids except for glc.

### 3.4. Polymerization during Extended Wet-Dry Cycling

To characterize oligomerization during extended wet-dry cycling, four mixtures were subjected to wet-dry cycling over 8 weeks (4 days in the dry state alternating with 3 days in the wet state). Mixtures of lac+Ala, lac+β-Aba, hba+Ala, and hba+β-Aba were characterized. Consistent with Forsythe et al. [45], MS analysis after the final cycle shows a trend of enrichment in oligomer products of amino acid over hydroxy acids compared to short-term wet-dry cycling (Compare Appendix A to Appendix A). For example, MS spectra analysis for lac+Ala shows shorter depsipeptides at the end of the extended cycle (maximal length of 11-m in the prolonged wet-dry cycling versus a 16-mer in the 7-day single-step dry-down reactions) with greater amino acid content (e.g., 4lac7Ala in Appendix A compared with 10lac5Ala in Appendix A). HPLC analysis of samples subjected to extended wet-dry cycling shows that the product distribution continuously shifted over the course of the cycling experiment, albeit approaching steady state after the 8th cycle for all four combinations tested (Appendix A). ^1^H NMR analysis indicated that the percent conversions of the amino acids after eight cycles were 40% for lac+Ala (Appendix A), 48% for lac+β-Aba (Appendix A), 26% for hba+Ala (Appendix A), and 26% for hba+β-Aba (Appendix A). It is evident that the percent conversion of β-Aba, but not of Ala, into products at the end of the 8th week was higher than that observed after daily wet-dry cycling for 1 week (Table 1). For example, β-Aba exhibited 26% conversion into oligomers when mixed with hba after wet-dry cycling for 8 weeks, whereas only 11% conversion was observed following daily wet-dry cycling for 1 week.

## 4. Discussion

We studied the formation of depsipeptides in condensation-dehydration reactions involving mixtures of alpha and beta hydroxy acids and amino acids. The goal was to provide insight into prebiotic chemical factors that influenced the origins of extant protein backbone structure. The reactions investigated here produce peptide bonds through hydroxy-acid-mediated ester-amide exchange, and are driven by low water activity without the need for activation by high-energy compounds. Our results demonstrate that both alpha and beta hydroxy acids catalyze peptide bond formation, resulting in the production of depsipeptides. Amino acid incorporation into depsipeptides is generally greater with alpha hydroxy acids than with beta hydroxy acids. The degree of polymerization by beta amino acids is generally greater than that of alpha amino acids in the same reactions. This report is the first demonstration that beta hydroxy acids can promote depsipeptide formation in single-step dry-down reactions. In the future it will be interesting to investigate whether these observations will also be evident in ‘one-pot’ reactions containing both alpha and beta amino acids, and to determine if selective patterns will emerge.

Several factors likely determine the relative reactivities of amino acids when combined with alpha hydroxy acids or with beta hydroxy acids. First, ester-linked alpha hydroxy acid homodimers and amide-linked alpha hydroxy acid-alpha amino acid heterodimers can cyclize into 6-membered rings, which can then participate in ring-opening polymerization (ROP). ROP is a common methodology for chain growth polymerization via the use of heterocyclic monomers, which have greater reactivity than their corresponding linear forms [64,65,66,67]. In the context of depsipeptide formation, ROP of dilactone rings (alpha hydroxy acid cyclic dimers) and morpholinediones (alpha amino acid-alpha hydroxy acid heterodimers) is expected to increase the susceptibility to nucleophilic attack of the ester-carbonyl to ester-amide exchange and ester–ester exchange [68]. Another possible contribution to the higher reactivities of amino acids combined with alpha hydroxy acids compared to with beta hydroxy acids is differences in the starting, unbuffered pH of the reactions reported here. The starting pH of solutions containing alpha hydroxy acids was lower (by ~0.3–0.4 pH units) than of solutions containing beta hydroxy acids. Lower pH is expected to promote more Fischer esterification, which would enrich the oligomers in hydroxy acids and in ester bonds.

Our determination that beta amino acids are more readily incorporated into depsipeptides than alpha amino acids, based on MS and NMR analysis, is consistent with the finding of Liu and Orgel that β-glutamic acid polymerizes more efficiently than its alpha analog in the presence of activating agent 1-ethyl-3-(3-dimethylaminopropyl)carbodiimide (EDAC) [55]. There are several possible explanations for the higher reactivities of beta amino acids compared to alpha amino acids observed in our system. First, the nucleophilicity of the amine group of beta amino acids is expected to be greater than that of alpha amino acids due to attenuation of electron withdrawal by the carboxylic acid in beta amino acids versus alpha amino acids. The carboxylic acids of beta amino acids, with higher pKa’s, are also better electrophiles than those of the alpha analogs. Indeed, various products were observed for beta amino acids in which the amino acid has been esterified with a hydroxy acid, while these products are not as abundant in reactions that contained alpha amino acids.

We also investigated effects of steric hindrance in amino and hydroxy acids on reactivities. Unhindered glc was compared to hindered lac and unhindered Gly was compared to hindered Ala. Similarly, unhindered hpa was compared to hindered hba and unhindered β-Ala was compared to hindered β-Aba. As expected, the methyl group adjacent to the nucleophilic amine/alcohol damped reactivity, confirmed by lower percentage conversions in methylated hydroxy or amino acid compared to their nonmethylated counterparts (Table 1).

We observed remarkable differences in the extent of incorporation of amino acids into oligomers when comparing reactions conducted under single-step dry-down reactions to those conducted under wet-dry cycling conditions. Because wet phases shift the driving force towards hydrolysis, we expected that higher extents of polymerization of amino acids would be observed under single-step dry-down reactions compared to those observed under wet-dry cycling. However, this was not always the case; in about one-third of the monomer combinations, a greater extent of polymerization was observed under wet-dry cycling conditions. The observed difference in extent of polymerization between single-step dry-down reactions versus wet-dry cycling implies that the systems do not reach equilibrium and that one or both of the systems are kinetically trapped. These results might imply that administration of water to the system might promote better diffusion or break crystal formation that can form during the drying process [69]. Moreover, these results highlight the significance of exploration of different reaction conditions with different mixture types, as different mixtures do not behave the same under dried conditions in regard to perturbation in hydration levels.

Finally, in a previous investigation we showed that the proteinaceous cationic amino acids Arg, His, and Lys oligomerize with higher efficiencies and regioselectivities than nonproteinaceous cationic analogs Orn, Dab, and Dpr [32], and generated depsipeptides that are superior in terms of stabilizing RNA structures compared to nonproteinaceous cationic analogs [35]. The results of those studies suggested that the greater reactivity of the proteinaceous cationic amino acids could have been a prebiotic selection pressure that contributed to their ultimate selection for coded protein synthesis. In contrast, the results presented in this study suggest that the selection for alpha backbones came at a later stage in evolution and did not arise from specific chemical activity of amino acids themselves.

## Figures and Tables

**Figure 1 life-12-00265-f001:**
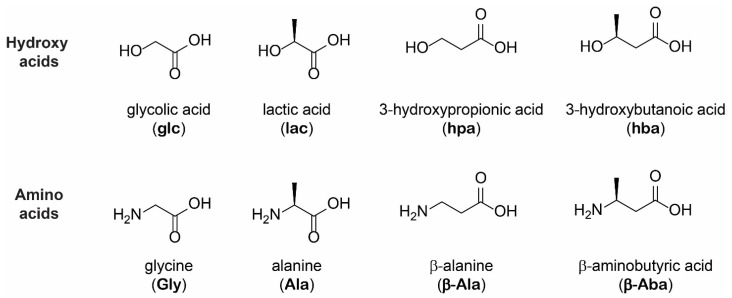
Chemical structures of alpha and beta hydroxy and amino acid monomers investigated in this study. Depsipeptides were generated by subjecting mixtures containing a single hydroxy acid along with a single amino acid to either single-step dry-down reactions or wet-dry cycling.

**Figure 2 life-12-00265-f002:**
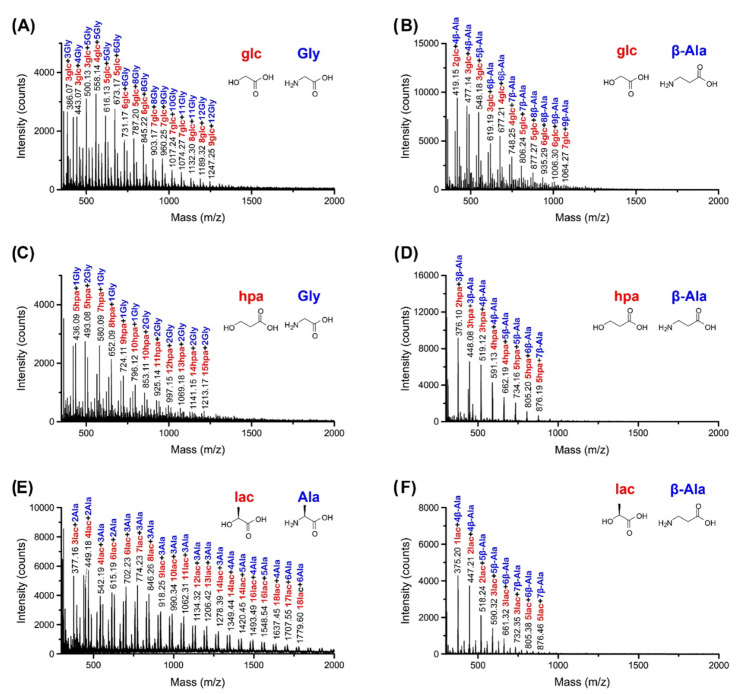
Mass spectrometry analysis of single-step dry-down reactions of alpha and beta hydroxy and amino acids support the formation of depsipeptides with various backbones. Glycolic acid (glc) and 3-hydroxypropionic acid (hpa) were subjected to a single-step dry-down with either glycine (Gly) (**A**,**C**, respectively) or β-alanine (β-Ala) (**B**,**D**, respectively) at 85 °C for seven days. Similarly, lactic acid (lac) was subjected to the same conditions with either Ala or β-Ala (**E**,**F**, respectively). The resulting oligomerization products were analyzed by MALDI-MS, indicating a variety of depsipeptides. glc, lac and hpa are labeled in red, Ala, Gly and β-Ala are labeled in blue. The product backbones depend on the composition of the starting mixture. Observed backbone products are all-alpha, all-beta, and heterogenous, containing both alpha and beta building blocks. Labeled species correspond to [M+H]^+^, [M+Na]^+^, or [M+K]^+^ ions.

**Figure 3 life-12-00265-f003:**
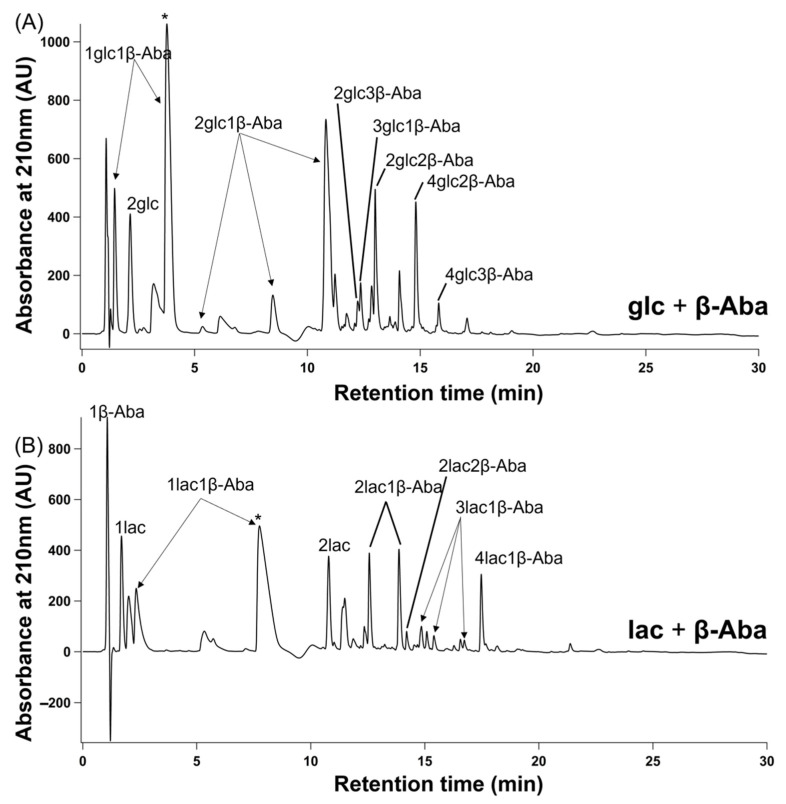
HPLC traces of single-step dry-down reactions of β-Aba. Mixtures of β-Aba with either glc (**A**) or lac (**B**) were subjected to a single-step dry-down at 85 °C for seven days. Asterisks indicate peaks that correspond to the sequences glcβ-Aba or lacβ-Aba, as verified via authentic standards.

**Figure 4 life-12-00265-f004:**
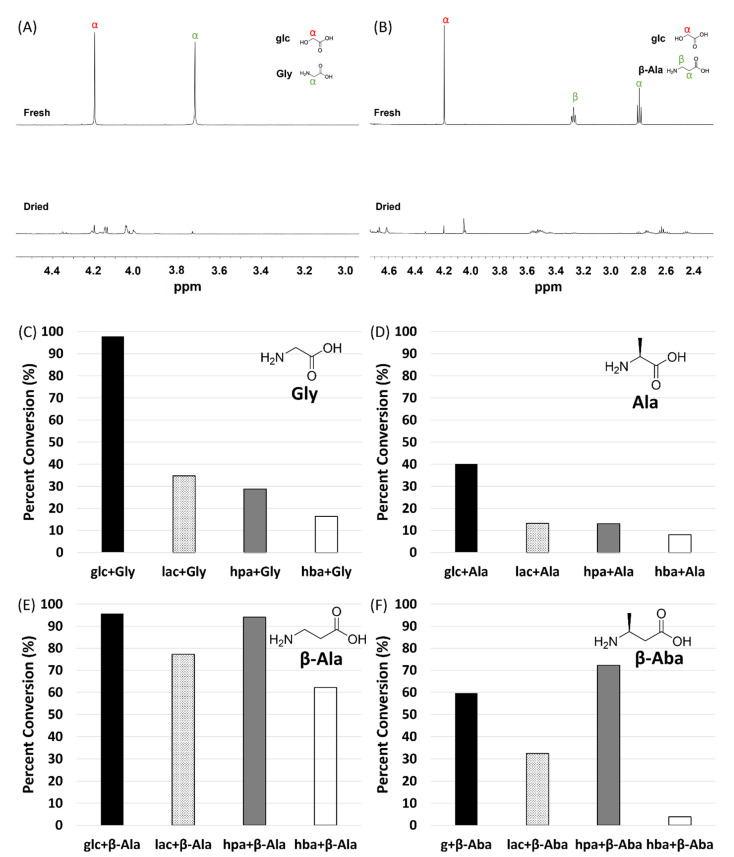
Comparison of ^1^H NMR spectra of single-step dry-down reactions (‘dried’) to non-dried (‘fresh’) mixtures allows determination of reactivities of alpha and beta amino and hydroxy acids. Mixtures of amino and hydroxy acids at 1:1 molar ratios were subjected to single-step dry-down reactions for 1 week at 85 °C. Spectra of mixtures of glc+Gly (**A**) or glc+β-Ala (**B**) in D_2_O before reaction (top trace) and after single-step dry-down reactions (bottom trace) are shown. The per cent conversion of Gly (**C**), Ala (**D**), β-Ala (**E**), or β-Aba (**F**) into products in the presence of all four hydroxy acids is shown.

**Figure 5 life-12-00265-f005:**
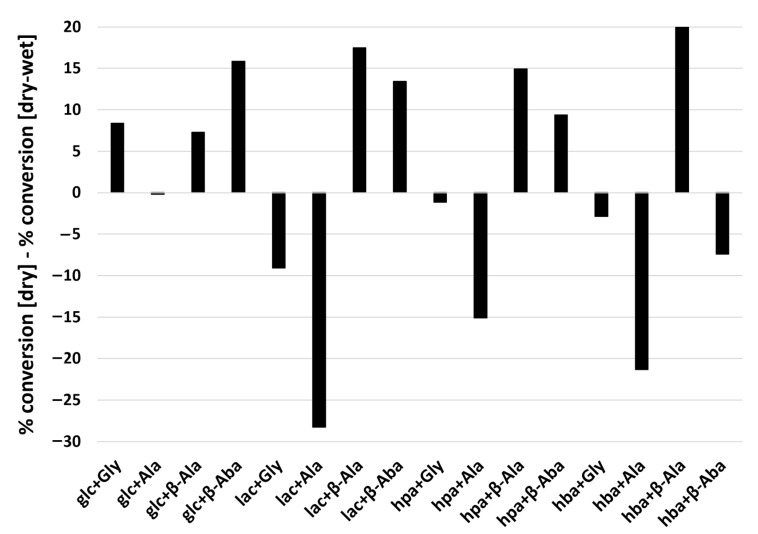
Differences in percent conversion of amino acid into products between single-step dry-down reactions versus wet-dry cycling. The graph shows results of subtractions of percent conversion of amino acids into products under single-step dry-down reactions from those under wet-dry cycling conditions. Positive values indicate an greater extent of conversion in single-step dry-down reactions whereas negative values indicate a lesser extent of conversion under wet-drycycling conditions. Raw values are shown in Table 1.

**Table 1 life-12-00265-t001:** Percent conversion of the four different amino acids into products in the various mixtures following single-step dry-down reactions or wet-dry cycling reactions.

Reaction	Overall Conversion—Single-Step Dry-Down (%) ^1^	Overall Conversion—Wet-Dry Cycling (%) ^1^	Reaction	Overall Conversion—Single-Step Dry-Down (%) ^1^	Overall Conversion—Wet-Dry Cycling (%) ^1^
glc+Gly	98	90	hpa+Gly	29	30
glc+Ala	40	40	hpa+Ala	13	28
glc+β-Ala	96	88	hpa+β-Ala	94	79
glc+β-Aba	60	44	hpa+β-Aba	72	63
lac+Gly	35	44	hba+Gly	16	19
lac+Ala	13	41	hba+Ala	8	30
lac+β-Ala	77	60	hba+β-Ala	62	40
lac+β-Aba	33	19	hba+β-Aba	4	11

^1^ Quantitated by integration of ^1^H NMR peaks. For integration of nonreacted amino acid resonances, either α- or β-protons were selected (for detailed information on the integrations and chosen chemical shifts, see Materials and Methods). Overall conversion refers to the conversion of an amino acid monomer into products.

## Data Availability

All data presented in this study are available in the manuscript itself and its Supporting Information.

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
