# Peer review of "Differential Oligomerization of Alpha versus Beta Amino Acids and Hydroxy Acids in Abiotic Proto-Peptide Synthesis Reactions"

_life, 2022, doi:10.3390/life12020265_

Round 1

Reviewer 1 Report

Differential oligomerization of alpha versus beta-amino acids and hydroxy acids in abiotic proto-peptide synthesis reactions

Frenkel-Pinter et al. Life

The author describes an experiment that compares the reactivities of alpha and beta-amino acids in the presence of hydroxy acids for the formation of depsipeptides under wet-dry protocol conditions. Though the importance of depsipeptides in synthesizing peptide chains in a prebiotic condition/protocol (wet-dry cycles, without activating agent) has previously been reported by the same lab (Forsythe et al. 2015 Angew. Chemie), the current study represents a significant advancement in understanding the chemical preferences of alpha and beta-amino acids under cyclic or single-step wet-dry protocols. The study is meticulously carried out, and the authors present strong analytics throughout the paper. After addressing a few minor concerns, I strongly recommend that the paper be published.

- One thing authors should clarify or be cautious about is the use of the term "selection" in the abstract, introduction, and discussion. Though the authors compare the efficiency of alpha vs beta-amino acids in their respective reactions, the experiment/s in which all of the reactants are present in a single-pot and then enrichment/selection (or chemical selection) of depsipeptides containing one type of aa over the other is not part of the current study. The results of a single-pot reaction could shed some light on chemical selection. Though I am not certain about the feasibility of such a reaction (and associated analytics), the author should address this in the manuscript (in the discussion section maybe).

-line 24: Control reactions were done only using the single wet-dry protocol. Do authors have data on the control reactions with cycling protocol? If yes, then it should be added to the manuscript or, if not, then can they please comment on it.

-Supplementary methods, synthesis of peptide standards (page 8-9): Authors should mention the quantities of substrates used in terms of ‘µmol’ and if yields are available for the reaction please mention that too.

Reviewer 2 Report

Frenkel-Pinter et. al., have investigated alpha versus beta amino acids and hydroxy acids polymerization reactions under prebiotic conditions. In general, it was an interesting piece of work in prebiotic chemistry. But the experiment design and some conclusions are not proper. I would not recommend to publish in Life in current form, but could be reconsidered if all major comments are addressed.

Major comments:

  1. 3-4 is not a general pH value under prebiotic condition. As CO2 dissolved in the ocean, the pH value of the early ocean can be 3-4. But the early ocean is impossible evaporated to dryness. The dry-wet cycle is more likely happened in a lake or pond. The pH value of lakes or ponds are variable (1-12). The authors should compare these reactions starting from different pH value.
  2. The conclusion in line 235-236, 402-403 is not supported by the data. MS can be used as a quantitative method ONLY IF the authors have all standard curves for every single sample. In current paper, there are not any standard curves, and it is impossible for authors to have all these curves. Thus, I have no idea how did the authors have the conclusions of ‘dominant species’ in table S1 as well. Apart from that, by counting the peaks of different products, comparing Fig 2A with Fig 2B, beta-alanine is not more enriched than glycine. Fig 2E and Fig 2F are not comparable, because beta-alanine is less hindered than alanine. Fig S6 (Fig 2E) can be compared with Fig S8 instead. But alanine is more incorporated than beta-aminobutyric acid. All these conclusions and the related sentences should be removed from the paper.
  3. The authors didn’t discuss about MS spectra isotope pattern in the paper, or use software to calculate the isotope pattern for each peak before assign the peaks. I would suggest the authors only assign the major peaks, that can make the figures much more readable than current form as well.
  4. The authors used the integration of 1H NMR peak to calculate the conversion yield (Table 1 and Fig 5). The authors should do a control reaction (only amino acids under the same condition) to prove the amino acids do not have any side reaction (decompose or precipitate out of the solution, etc.).
  5. Fig S71 and S72 are wrong spectra which doesn’t have hba in the spectra.

Minor comments:

  1. Entries should be added to the tables, and cited in the manuscript accordingly.
  2. Label A and B are missing in Fig S18.

Reviewer 3 Report

The paper offers an interesting and nice piece of chemistry, reporting a systematic comparison of the reactivities of alpha versus beta amino and hydroxy acids in depsipeptide polymerization reactions in single-step dry-down or dry/wet cycling reactions. Surprisingly, the authors omitted to quote their previous paper describing depsipeptides that form under mild dry-down reactions (Frenkel-Pinter M et al. 2019, Selective incorporation of proteinaceous over non proteinaceous cationic amino acids in model prebiotic oligomerization reactions. PNAS 116:6338-16346).

The authors should point the differences between the two papers.

The authors should also clearly mention previous experiments showing selective polymerization:

  • Zhao Y-F, Cao P-S (1999) Why nature chose alpha-amino acids. Pure Appl Chem 71: 1163-1166.

As well as some general considerations:

  • Weber AL, Miller SL (1981) Reasons for the Occurrence of the Twenty Coded Protein Amino Acids. J Mol Evol 17:273-284).

Reviewer 4 Report

The authors report about a careful and very detailed investigation of the condensation of different types of amino acids and hydroxy acids in potentially prebiotic settings. The reactions were analyzed by mass spectrometry and NMR. The results obtained are of interest to those working in the field of prebiotic amino acid and peptide chemistry. The manuscript is well written. Follow-up investigations could be on the possible influence of interfaces (solid or soft) on some of the reactions investigated and on the properties of some of the products obtained, for example in terms of protocell membrane permeabilization. The question is whether the formation of the oligomers leads to some emergent properties that are not present in the building blocks from which the oligomers formed.

Publication of the manuscript in Life is recommended. There is only comment:

Figure 2: The data shown do not fit with the figure legend. Please check and correct!

Round 2

Reviewer 2 Report

The authors should make a note about the reason of absence of hba in Fig S71 and S72, just in case the readers are puzzled. Because hba was clearly presence in Fig S73 and 74.

Author Response

Reviewer 2, Comment 1: The authors should make a note about the reason of absence of hba in Fig S71 and S72, just in case the readers are puzzled. Because hba was clearly presence in Fig S73 and 74.

Response: We added a note in the legend of Figures S71-S72.
